# The Bioactive Gamma-Oryzanol from *Oryza sativa* L. Promotes Neuronal Differentiation in Different In Vitro and In Vivo Models

**DOI:** 10.3390/antiox13080969

**Published:** 2024-08-09

**Authors:** Giulia Abate, Alex Pezzotta, Mariachiara Pucci, Valeria Bortolotto, Giovanni Ribaudo, Sara A. Bonini, Andrea Mastinu, Giuseppina Maccarinelli, Alberto Ongaro, Emanuela Tirelli, Daniela Zizioli, Alessandra Gianoncelli, Maurizio Memo, Mariagrazia Grilli, Daniela Uberti

**Affiliations:** 1Department of Molecular and Translational Medicine, University of Brescia, 25123 Brescia, Italy; giulia.abate@unibs.it (G.A.); giovanni.ribaudo@unibs.it (G.R.); sara.bonini@unibs.it (S.A.B.); andrea.mastinu@unibs.it (A.M.); giuseppina.maccarinelli@unibs.it (G.M.); a.ongaro005@unibs.it (A.O.); e.tirelli004@unibs.it (E.T.); daniela.zizioli@unibs.it (D.Z.); alessandra.gianoncelli@unibs.it (A.G.); maurizio.memo@unibs.it (M.M.); daniela.uberti@unibs.it (D.U.); 2Department of Medical Biotechnology and Translational Medicine, University of Milan, 20133 Milan, Italy; alex.pezzotta@unimi.it; 3Laboratory of Neuroplasticity, University of Piemonte Orientale, 28100 Novara, Italy; valeria.bortolotto@uniupo.it (V.B.); mariagrazia.grilli@uniupo.it (M.G.); 4Department of Pharmaceutical Sciences, University of Piemonte Orientale, 28100 Novara, Italy

**Keywords:** gamma-oryzanol, phytocomplex, natural Nrf2 inducers, neural progenitor cells, zebrafish model, molecular docking

## Abstract

Gamma-oryzanol (ORY), found in rice (*Oryza sativa* L.), is a mixture of ferulic acid esters with triterpene alcohols, well-known for its antioxidant and anti-inflammatory properties. Our past research demonstrated its positive impact on cognitive function in adult mice, influencing synaptic plasticity and neuroprotection. In this study, we explored whether ORY can exert neuro-differentiating effects by using different experimental models. For this purpose, chemical characterization identified four components that are most abundant in ORY. In human neuroblastoma cells, we showed ORY’s ability to stimulate neurite outgrowth, upregulating the expression of GAP43, BDNF, and TrkB genes. In addition, ORY was found to guide adult mouse hippocampal neural progenitor cells (NPCs) toward a neuronal commitment. Microinjection of ORY in zebrafish Tg (*-3.1 neurog1*:GFP) amplified *neurog1*-GFP signal, *islet1*, and *bdnf* mRNA levels. Zebrafish *nrf2a* and *nrf2b* morphants (MOs) were utilized to assess ORY effects in the presence or absence of Nrf2. Notably, ORY’s ability to activate *bdnf* was nullified in *nrf2a-MO* and *nrf2b-MO*. Furthermore, computational analysis suggested ORY’s single components have different affinities for the Keap1-Kelch domain. In conclusion, although more in-depth studies are needed, our findings position ORY as a potential source of bioactive molecules with neuro-differentiating potential involving the Nrf2 pathway.

## 1. Introduction

Gamma-oryzanol (ORY) is a naturally occurring compound found in rice (*Oryza sativa* L.), consisting of at least ten well-characterized phytosteryl ferulates, with the major components accounting for approximately 80% being cycloartenyl, 24-methylenecycloartanyl, campesteryl, and sitosteryl ferulates [1]. For several years, extensive research has been conducted to demonstrate the health benefits of ORY, which include the prevention of chronic diseases and the promotion of healthy brain aging [2].

Emerging findings from various research groups, including our own, have demonstrated that ORY exerts neuroprotective effects and improves cognitive performance [2,3,4]. Notably, our previous study showed that chronic oral administration of ORY is able to prevent behavioral alterations and neuronal inflammation induced by lipopolysaccharide (LPS) in adult mice [3]. Employing a proteomic approach, we observed that mice chronically treated with ORY showed increased expression of proteins associated with synaptic plasticity, mitochondrial energy metabolism, and neuroprotection, resulting in improved cognitive performance [4].

From a mechanistic point of view, ORY exerts its antioxidant effect by scavenging free radicals and activating the Nuclear factor erythroid 2-related factor 2 (Nrf2) pathway [1]. Nrf2 is a transcription factor that plays a critical role in regulating the expression of genes involved in the antioxidant response and detoxification pathways. Under physiological conditions, Nrf2 is inactive in the cytoplasm due to its binding to Keap1. However, when cells encounter oxidative stress or other harmful factors, Nrf2 is released from Keap1 and translocated to the nucleus, where it binds to specific DNA sequences known as antioxidant response elements (AREs). These events lead to the activation of genes responsible for antioxidant and cytoprotective functions [5].

More recently, crosstalk between Nrf2 and brain-derived neurotrophic factor (BDNF) has emerged [6,7]. It has been shown that activation of Nrf2 can induce the expression of BDNF in neurons, and that BDNF can in turn activate Nrf2 and increase antioxidant gene expression [8]. Interestingly, Nrf2 knockout mice exhibited significantly reduced levels of BDNF, and a Nrf2 consensus sequence was found within the promoter region for the BDNF gene [9,10]. Considering the pivotal roles of BDNF in synaptic plasticity and cognitive abilities, along with Nrf2’s protective role in reducing oxidative stress and inflammation, compounds able to act on these key players hold promise for the treatment and prevention of various neurological disorders [6,8].

Hence, in this study, our aim will be to explore whether ORY, as an Nrf2 activator, can exert neuro-differentiating potential in different experimental models. Starting from a phytochemical characterization of ORY, the investigation will extend to human neuroblastoma cells and adult mouse neural progenitor cells (NPCs), providing a comprehensive assessment of ORY’s influence on neurite outgrowth, BDNF modulation, and NPC neuronal commitment, respectively. Then, experiments involving *nrf2a-* and *nrf2b*-deficient zebrafish will be conducted to demonstrate Nrf2’s role in embryonic neuronal development. Finally, an in silico approach will be used to identify which molecules within the ORY phytocomplex can strongly interact with Keap 1, potentially influencing Nrf2 pathway activation.

## 2. Materials and Methods

### 2.1. HPLC Characterization

ORY (Sigma Aldrich, Merck KGAA, Darmstadt, Germany) was characterized via high-performance liquid chromatography (HPLC) using a Dionex UltiMate 3000 system (Thermo Fisher Scientific, Waltham, MA, USA). ORY was dissolved in 2-propanol and injected into a 5 μm, 250 × 4.6 mm, 100 Å Kinetex Evo C18 column (Phenomenex, Torrance, CA, USA) pre-equilibrated with a mobile phase solution of a 15:85 mixture of methanol and acetonitrile. The LC parameters were as follows: mobile phase composition of 15:85 methanol and acetonitrile, flow rate of 0.3 mL/min, and analysis temperature of 40 °C. Detection was performed at 323 nm. Compound identity was confirmed using mass spectrometry (MS) analysis with an LCQ Fleet Ion Trap MSn electrospray ionization (ESI) spectrometer (Thermo Fisher Scientific, Waltham, MA, USA) in negative ionization mode (m/z 100–2000). ESI conditions were as follows: source voltage of 5.3 kV, capillary voltage of −16.00 V, capillary temperature of 350 °C, and nitrogen sheath (35 arb) and auxiliary (15 arb) gas. Data analysis was conducted using Xcalibur software 4.0.27.13 (Thermo Fisher Scientific, Waltham, MA, USA). A representative mass spectrum displaying identified molecular ion peaks is provided in Appendix A.

### 2.2. Cell Culture and Immunofluorescence Analysis

Human neuroblastoma cells were purchased from the American Type Culture Collection (ATCC, Manassas, VA, USA). These cells are cryopreserved in the ISO 9001:2015-certified cryogenic bank at the University of Brescia, where they are also cultivated and used only for research purposes. Human neuroblastoma cells (SH-SY5Y) were cultured in a 1:1 ratio of Ham’s F12 medium and Dulbecco-modified Eagle’s medium (DMEM, Sigma Aldrich), supplemented with fetal bovine serum (10%, Sigma Aldrich, Merck KGAA, Darmstadt, Germany) and L-glutamine (0.5%, Sigma Aldrich) at 37 °C, 5% CO_2_ until 80% confluence. Cells were seeded at a density of 5 × 10^4^ cells/cm^2^ on poly-D-lysine-coated glass coverslips in 24-well plates. After seeding, cells were treated with: (i) Retinoic acid (RA) at 10 µM and (ii) three doses (1, 5, and 10 µg/mL) of ORY as per a prior study [1]. After 5 days, cells were harvested for gene and protein analysis or fixed in methanol for immunofluorescence. Neuronal morphology was visualized by immunolabeling with anti-βIII-tubulin antibody and DAPI counterstaining. Cells were incubated with primary anti-βIII tubulin (1:200, Chemicon Millipore) in PBS containing 1% BSA and 0.2% Triton X-100 overnight at 4 °C, followed by Cy3 (1:500, Invitrogen, Buenos Aires, Argentina) conjugated secondary antibody for 1 h at room temperature. Cell nuclei were stained with DAPI (1:3000, Sigma Aldrich) for 3 min. Confocal microscopy (LSM 900 META, Carl Zeiss S.p.A., Milan, Italy) was used for fluorescence imaging (1024 × 1024 pixels) and image reconstruction was performed with an LSM Zen Blue Image Examiner (Carl Zeiss S.p.A., Milan, Italy). The percentage of morphologically differentiated cells was determined manually by analyzing at least 10 fields for each treatment; cells with neurites ≥ 50 μm in length were considered differentiated [11].

### 2.3. Western Blot Analysis

Cells were harvested in 80 µL of lysis buffer containing 50 mM Tris-HCl (pH 7.6), 150 mM NaCl, 5 mM EDTA, 1 mM phenyl methyl sulphonyl fluoride, 0.5 mg/mL leupeptin, 5 mg/mL aprotinin, and 1 mg/mL pepstatin. Protein content was determined by a conventional method (BCA protein assay kit, Pierce, Rockford, IL, USA). Fifty micrograms of protein extracts were electrophoresed on 15% SDS-PAGE and transferred to nitrocellulose paper (Schleicher and Schuell, Dassel, Germany). Membranes were blocked for 1 h in 3% w/v Bovine Serum Albumin in TBS-T (0.1 M Tris-HCl pH 7.4, 0.15 M NaCl, 0.1% Tween 20) and incubated overnight at 4 °C with primary antibodies. Primary antibodies were anti-GAPDH (1:2500, Sigma-Aldrich) and anti-BDNF (1:200, Santa Cruz Biotechnology, Dallas, TX, USA). IR Dye near-infrared dye-conjugated secondary antibodies (LI-COR, Lincoln, NE, USA) were used. Immunodetection was performed using a dual-mode Western imaging system, Odyssey FC (LI-COR Lincoln, NE, USA). Quantification was performed using Image Studio Software 3.1 (LI-COR, Lincoln, NE, USA) and the results were normalized over the GAPDH signal.

### 2.4. Subcellular Fractionation for Nrf2 Nuclear Translocation

Nuclear protein extracts were prepared by washing SH-SY5Y cells treated and untreated with ORY (10 µg/mL) twice with ice-cold PBS. Cells were subsequently homogenized 15 times using a glass–glass dounce homogenizer in 0.32 M sucrose buffered with 20 mM Tris hydrochloride (Tris-HCl) (pH 7.4) containing 2 mM ethylenediaminetetraacetic acid (EDTA), 0.5 mM ethylene glycol-bis(2-aminoethylether)-N,N,N′,N′-tetraacetic acid (EGTA), 50 mM β-mercaptoethanol, and 20 μg/mL leupeptin, aprotinin, and pepstatin. The homogenate was centrifuged at 300 g for 5 min to obtain the nuclear fraction. An aliquot of the nuclear fraction was used for protein assays conducted using the Bradford method, whereas the remaining was boiled for 5 min after dilution with sample buffer and subjected to polyacrylamide gel electrophoresis and immunoblotting for the detection of Nrf2. Briefly, 25 µg of nuclear extract were electrophoresed on 12% SDS-PAGE and transferred to nitrocellulose paper (Schleicher and Schuell, Dassel, Germany). Membranes were blocked for 1 h in 3% *w*/*v* Bovine Serum Albumin in TBS-T (0.1 M Tris-HCl pH 7.4, 0.15 M NaCl, 0.1% Tween 20) and incubated overnight at 4 °C with primary antibodies. Primary antibodies were anti-Nrf2 (1:300, Santa Cruz Biotechnology, Dallas, TX, USA) and anti-Histone H3 (1:1000, Cell Signaling, Danvers, MA, USA). IR Dye near-infrared dye-conjugated secondary antibodies (LI-COR, Lincoln, NE, USA) were used. Immunodetection was performed using a dual-mode Western imaging system, Odyssey FC (LI-COR Lincoln, NE, USA). Quantification was performed using Image Studio Software 3.1 (LI-COR, Lincoln, NE, USA) and the results were normalized over the Histone H3 signal.

### 2.5. Quantitative Real-Time PCR and Gene Expression

SH-SY5Y-treated cells were analyzed for mRNA expression. Total mRNA was extracted with TRI Reagent (Sigma Aldrich, Merck KGAA, Darmstadt, Germany), reverse transcribed with M-MLV reverse transcriptase (Promega, Madison, WI, USA), and cDNA was used for qRT-PCR. Human-specific primers for GAP-43, BDNF, TrkB, and GAPDH are reported in Table 1 (Metabion international AG, Planegg, Germany). The ViiA7 Real-Time PCR Detection System (Applied Biosystems, Waltham, Massachusetts, USA) was used for amplification with the following conditions: 10 min at 95 °C, followed by 40 cycles of 1 s at 95 °C and 30 s at 60 °C. GAPDH was used as the reference gene, and the fold change in target gene expression was calculated using the comparative Ct method [12].

### 2.6. Animals

Male wild-type (WT; C57BL/6) mice were obtained from Jackson Laboratories (Bar Harbor, ME, USA) and housed in HEPA-filtered Thoren Units (Thoren Caging System Inc., Hazleton, PA, USA) with 3–4 animals per cage and provided with ad libitum food and water at the University of Piemonte Orientale animal facility. Animal care and experimental procedures were conducted in accordance with the European Community Directive and approved by the Institutional Animal Care and Use Committees (IACUC) at the University of Piemonte Orientale, Italy. The ethical approval number is DB064.47.

### 2.7. Culture of Adult Murine Hippocampal Neural Progenitor Cells

Neural progenitor cells (NPCs) were isolated from the hippocampi of three 3–4-month-old male mice and processed as previously described. NPCs, grown as floating neurospheres, were dissociated for the first time (passage 1, P1) after 7–10 days in vitro (DIV). Thereafter, cells were dissociated every five DIVs. After each dissociation, cells were kept in culture at a density of 12,000 cells/cm^2^ in complete culture medium composed of Neurobasal-A medium supplemented with 2% B27, 2 mM L-glutamine (Gibco, Life Technologies, Monza, Italy), recombinant human epidermal growth factor (rhEGF, 20 ng/mL), recombinant human fibroblast growth factor 2 (rhFGF-2, 10 ng/mL; Peprotech, Rock Hill, NJ, USA), heparin sodium salt (4 µg/mL; Sigma-Aldrich), 100 U/mL penicillin, and 100 µg/mL streptomycin (Gibco^TM^ Thermo Fisher Scientific, Waltham, MA, USA). For experimental procedures, NPCs were utilized at passages P2–P30 from at least two distinct cell preparations.

### 2.8. Neural Progenitor Cell Differentiation

For differentiation assessment, a previously described protocol was utilized [13]. Briefly, NPCs were plated at a density of 43,750 cells/cm^2^ on laminin-coated (2.5 µg/cm^2^) Lab-Tek 8-well Permanox chamber slides (NUNC, Thermo Fisher Scientific, Waltham, MA, USA) in differentiation medium. NPCs were differentiated for 24 h in the presence of a vehicle (0.2% DMSO) or ORY at different concentrations (1, 5, and 10 µg/mL). After, NPCs were fixed with ice-cold 4% paraformaldehyde/4% sucrose solution for 20 min at room temperature and processed by immunocytochemical analysis as previously described [13]. Briefly, cells were incubated overnight at 4 °C with the following primary antibodies: anti-microtubule-associated protein-2 (MAP-2, rabbit polyclonal, 1:600, Millipore, Milan, Italy) as a marker for mature neurons and anti-glial fibrillary acidic protein (GFAP, mouse polyclonal, 1:600, MilliporeSigma, Merck KGAA, Darmstadt, Germany) as a marker for astrocytes. Thereafter, cells were incubated for 2 h at room temperature with the following secondary antibodies: Alexa Fluor 555-conjugated goat anti-rabbit IgG (1:1400, Molecular Probes, Life technologies) and Alexa Fluor 555-conjugated goat anti-mouse IgG (1:1600, Molecular Probes, Life technologies). Quantification was assessed as previously reported [14]: briefly, in each experiment, 150–200 cells corresponding to about five fields/well were counted using a DMIRB fluorescence microscope (Leica, Wetzal, Germany) with a 60× objective. Immunopositive cells were quantified, and their percentage was calculated over total viable cells. Nuclei were counterstained with 0.8 ng/mL Hoechst 33342 (Thermo Fisher Scientific, Waltham, MA, USA). Apoptotic nuclei were identified by Hoechst staining and counted, and the percentage of apoptotic cells over the total cell number was calculated.

### 2.9. Zebrafish Maintenance

Zebrafish embryos were raised and maintained according to the national guidelines (Italian decree March 4, 2014, n. 26) and the standard rules have been defined by the Local Committee for Animal Health. Embryos of the Tg *(-3.1neurog1*:GFP) [15] strain were collected by natural spawning, staged according to the reference guidelines, and raised in 28.5 °C fish water (instant ocean, 0.1% methylene blue in Petri dishes). From the stage of 24 h post-fertilization (hpf), 0.003% 1-phenyl-2-thiourea (PTU; Merck KGaA, Darmstadt, Germany) was added to prevent pigmentation. Prior to manipulation, embryos were anesthetized with 0.016% tricaine (ethyl 3-aminobenzoate methanesulfonate salt; Merck KGaA, Darmstadt, Germany).

### 2.10. Morpholino Microinjection

*nrf2a-MO* (5′-CATTTCAATCTCCATCATGTCTCAG-3′) and *nrf2b-MO* (5′-AGCTGAAAGGTCGTCCATGTCTTCC-3′), designed by Gene Tools LLC (Philomath, OR, USA), have previously been validated, as described in the literature [16]. MOs were injected into zebrafish embryos at cell stage one in parallel with a control MO (St-MO) according to the Gene Tool guidelines. A dose curve with an ascendent dose of both morpholinos has been established in order to find the optimal dose that we selected as follows: 4 nl of a solution containing the phenol red tracer, water, and the proper MO at the concentration of 0.3 pmol/e (pmol/embryo). After injection, embryos were then transferred in fish water containing 0.003% 1-Phenyl-2-thiourea (PTU) at 28 °C to stop the pigmentation process, and development was evaluated at 24 and 48 hpf.

### 2.11. Gamma-Oryzanol Microinjection and Fluorescence Evaluation

Dechorionated Tg (*-3.1neurog1*: GFP) embryos, at the stage of 1 day post-fertilization (dpf), were anesthetized and placed into Petri dishes for the microinjection of 100 µg ORY or DMSO in the hindbrain ventricle. After injection, embryos were incubated at 28.5 °C in fish water containing PTU. At the stage of 2 dpf, images of the head region of injected embryos were acquired under fluorescence microscopy in two independent biological replicates (Leica, Wetzlar, Germany; MZ FLIII). Images were analyzed through the use of ImageJ software 3.1 (National Institutes of Health, Bethesda, MD, USA) for the evaluation of the relative fluorescence intensity.

### 2.12. Whole Mount In Situ Hybridization (WISH)

First, 2 dpf zebrafish embryos of the Tg (*-3.1neurog1*: GFP) line were fixed overnight in 4% paraformaldehyde (Sigma-Aldrich) in Phosphate Buffer Saline (PBS) at 4 °C, then processed according to the protocol described by Thisse and Thisse [17]. The antisense *islet1* riboprobe was previously in vitro labelled with modified nucleotides (i.e., digoxigenin, Roche Diagnostics, Monza, Italy). Images of the *islet1* relative signal were taken under stereomicroscopy for two biological replicates (Leica, Wetzlar, Germany). According to the relative intensity of the signal, embryos were then divided into different categories: strong, normal, or reduced.

### 2.13. Computational Studies

The macromolecular target structure was obtained from the RCSB Protein Data Bank (PDB ID 4XMB). PDB file selection followed procedures outlined in prior works [18]. Ligand 3D models were constructed using Avogadro 1.2.0 and optimized with the same software [19]. Blind docking experiments were performed using AutoDock Vina (Molecular Graphics Laboratory, Department of Integrative Structural and Computational Biology, The Scripps Research Institute, San Diego, CA, USA) [20]. Prior to docking simulations, chain A of the PDB file was isolated and other molecules were removed. The receptor search volume was set with the following grid parameters to encompass the entire protein: Grid center: x = −6.634, y = 5.225, z = −13.876; size: 40 × 40 × 40 Å. The number of docking poses was set to 10, with other Vina parameters set as default. Residue numbering used in the PDB file was adopted. Output data, such as calculated binding energies and interaction patterns, were processed using the UCSF Chimera molecular viewer [21], which was also used to produce the artworks. Calculated binding energy values are expressed in kcal/mol and refer to the most favored predicted pose.

### 2.14. Statistical Analysis

Statistical analysis was performed using GraphPad Prism version 8. Ordinary One-way ANOVA tests with the Bonferroni post-test were used for neuronal cell studies, and data were presented as means ± S.E.M. Zebrafish assay data were analyzed with One-way ANOVA with Tukey correction and presented as means ± S.E.M. Violin box plots were used for graphical representation. In NPC experiments, data were expressed as means ± SD and analyzed by GraphPad Prism 7.0 using a two-tailed Student’s t-test, with *p*-values < 0.05 considered significant. Each experiment was performed three times in triplicate.

## 3. Results

### 3.1. Chemical Characterization of the Gamma-Oryzanol Mixture

HPLC characterization of ORY mixture compounds identified the presence of the four most abundant components, which are cycloartenyl ferulate (r.t. 8.14 min), 24-methylenecycloartanyl ferulate (r.t. 9.01 min), campesteryl ferulate (r.t. 9.56 min), and β-sitosteryl ferulates (r.t. 10.86 min) (Figure 1). Details in the ESI-MS spectrum of the four ORY components are further reported in Appendix A.

### 3.2. Gamma-Oryzanol Promotes Neuronal Differentiation in SH-SY5Y Neuroblastoma Cells

In vitro experiments were performed by treating human SH-SY5Y neuroblastoma cells with various concentrations of ORY mixture (1, 5, and 10 µg/mL) for 5 days. Additionally, a 5-day treatment with Retinoic Acid (RA, 10 µM) was included as a positive control for neuronal differentiation. As observed in Figure 2, RA treatment resulted in decreased cell proliferation and a significant increase in cells with neurite outgrowth (*p* < 0.001 ***). Surprisingly, ORY showed a dose-dependent induction of neuronal differentiation, as evidenced by β III tubulin staining, reducing proliferating cells and significantly increasing differentiated cells (Figure 2A–C). This effect was particularly notable at the highest concentration (CTRL vs. ORY 1 µg/mL *p* < 0.05 *; ORY 5 µg/mL *p* < 0.01 **; ORY 10 µg/mL *p* < 0.0001 ****).

The gene expression of Growth Associated Protein 43 (GAP43), involved in axon outgrowth and guidance, Brain-derived Neurotrophic Factor (BDNF), and its receptor tropomyosin receptor kinase B (TrkB) were evaluated. Both RA and ORY at 1 and 5 µg/mL significantly increased GAP43 gene expression after 5 days (Figure 3A). Similarly, BDNF and TrkB mRNA levels were enhanced after 5 days of ORY treatment, although the increase was not as substantial as observed with RA (Figure 3B,C). Notably, ORY at 10 µg/mL, despite not being statistically significant, showed higher levels compared to the control. Therefore, a dose-dependent, inverse relationship in the gene expression of BDNF, TrkB, and GAP43 was observed, suggesting that the temporal dynamics from gene expression to protein levels are crucial. To verify this, BDNF protein levels were assessed by Western blotting at the end of the treatment. The data confirmed that BDNF protein levels increased, following a pattern opposite to the initial gene expression (Appendix A).

### 3.3. Gamma-Oryzanol Promotes Neuronal Differentiation in Murine Adult Neural Progenitor Cells

To further confirm the differentiating effects of ORY, mouse adult neural progenitor cells from the hippocampus (ahNPCs) were engaged.

In particular, ahNPCs, grown in suspension as neurospheres, were allowed to differentiate through the removal of growth factors. After 24 h, ahNPCs gave rise to different subpopulations towards either the neuronal or glial phenotype, as established by MAP2- and GFAP-positive immunostaining, respectively. Under these experimental conditions, we also evaluated the effects of ORY at different concentrations. As reported in Figure 3A, ORY promoted neuronal differentiation of ahNPCs. In particular, 5 and 10 µg/mL ORY significantly increased the % of MAP2^+^ cells compared with vehicle-treated ahNPCs (Figure 4A,D,E), without affecting their survival (Figure 4C). Conversely, evaluation of ahNPC differentiation toward the astroglial lineage demonstrated that the % of GFAP^+^ cells was not affected by ORY treatment at lower concentrations (1 and 5 µg/mL), while at the highest dosage (10 µg/mL), ORY significantly reduced their percentage (Figure 4B,F,G).

### 3.4. Gamma-Oryzanol Exerts Pro-Neural Activity in the Presence of Functional Nrf2 Signalling in Zebrafish

Zebrafish possess two orthologues of the human NRF2 gene named *nrf2a* and *nrf2b* [22]. *nrf2* gene downregulation was assessed through the injection of antisense oligo morpholinos (MO; *nrf2a*-MO, *nrf2b*-MO) into Tg (*-3.1neurog1*:GFP) embryos, a transgenic line allowing in vivo visualization of the nervous system [16,22].

At the 1 dpf stage, these zebrafish models, along with control embryos injected with the same amount of control morpholino (std-MO), were injected in the hindbrain ventricle with DMSO or ORY (100 µg). The dose setting for ORY injection is detailed in Appendix A. One day later, they were analyzed for relative fluorescence intensity quantification. Compared to std-MO DMSO control embryos (Figure 5A,G), downregulation of *nrf2a* and *nrf2b* did not significantly reduce fluorescence intensity (Figure 5B,C,G). ORY injection increased fluorescence intensity in std-MO control embryos (Figure 5D,G).

However, the impaired Nrf2 axis due to *nrf2* gene downregulation did not increase the fluorescence signal after ORY injection (Figure 5E–G). To further strengthen these results, whole mount in situ hybridization was used to evaluate the relative expression of *islet1*, a marker of neuronal differentiation [23]. Downregulation of *nrf2a* and *nrf2b* resulted in a slightly reduced *islet1* signal compared to std-MO DMSO control embryos (Figure 5H–J,N). Consistent with fluorescence assay data, ORY increased *islet1* expression only in std-MO control embryos, not in embryos with *nrf2a* and *nrf2b* downregulation (Figure 5K–N).

To gain insight into the molecular mechanism by which ORY might exert its pro-neural action, we evaluated the levels of *bdnf* in the embryos injected with ORY. Additionally, in the same samples, we also investigated the expression levels of *gstp1* as a positive control of the injection, since its expression is modulated by ORY in the presence of a functional Nrf2 axis. As expected, ORY injection increased *gstp1* and *bdnf* expression only in std-MO control embryos, with no significant differences in embryos with downregulated *nrf2a* and *nrf2b* genes (Figure 5O,P). Altogether, these data indicate that ORY promotes neuronal development and differentiation by increasing BDNF production in the presence of functional Nrf2 signaling.

### 3.5. Computational Investigation of the Mechanism of Action of Key Compounds in the ORY Mixture

After verifying that ORY induces Nrf2 nuclear translocation in SHSY5Y cells as early as 3 h (Appendix A), a molecular docking experiment was conducted to better elucidate the mechanism by which the ORY mixture activates the Nrf2 pathway. In particular, we focused on the prediction of the binding mode of the four compounds contained in ORY with Keap1. In this connection, the X-ray structure of the Kelch domain from human Keap1 (PDB ID 4XMB) was selected as the macromolecular target for the considered compounds [24], in agreement with recent modeling studies on Nrf2 activators [18,25]. A blind docking approach was followed, and the conformational search space was expanded to encompass the whole target protein. Ferulic acid was introduced as a reference compound as ferulic acid is a chemical motif present in ORY components. The results of the docking study are reported in Figure 6. The calculated binding energy values were very promising for all ORY compounds (Figure 5B), and 24-methylenecycloartanyl ferulate was highlighted as the best molecule of the set (−11.1 kcal/mol). Interestingly, all ORY mixture compounds outperformed ferulic acid (−6.7 kcal/mol). In Figure 6C, a detailed view of the binding site is depicted, and the involved residues have been labeled. As can be expected, due to its molecular structure, 24-methyl cycloartanyl ferulate mainly interacts with hydrophobic residues (Ala, Gly, Ile, Leu, and Val) within the cavity. Interestingly, the compound, which is the most abundant in the ORY complex, as determined by the HPLC analysis carried out in the current study, was also predicted to be the strongest interactor of the set according to the docking score.

## 4. Discussion

In this study, by employing both in vitro and in vivo models, we established that the ORY mixture induces neuronal differentiation and neuronal development through the involvement of BDNF and the Nrf2 pathway.

We recently demonstrated in an in vitro model that ORY activated the Keap1–Nrf2 system through Nrf2 nuclear translocation and the transactivation of ARE-phase II antioxidant genes, HO1 and NQO1 [1]. These genes belong to a series of well-known NRF2 target genes that play pivotal roles in defense mechanisms, maintaining cellular redox homeostasis and facilitating detoxification from xenobiotics [26]. On the other hand, recent studies have postulated a novel function of Nrf2 in activating neurotrophic signaling pathways [27,28], and novel Nrf2 target genes involved in the regulation of cell fate determination have been identified [29].

Here, we demonstrated that ORY alone promoted neuronal differentiation of SH-SY5Y cells in a concentration-dependent manner. Such an effect is also supported by morphological changes and the expression of neuroactive players, such as GAP43 and BDNF. Notably, Zhao et al. [30] demonstrated that the Nrf2 activation pathway is essential for retinoic acid (RA)-induced SH-SY5Y differentiation. Evidence of such effects derived from experiments shows that the stable expression of Nrf2, achieved through transient lentivirus infection containing Nrf2 or chemical induction of endogenous Nrf2, promotes neuronal differentiation [30]. Conversely, inhibiting endogenous Nrf2 expression through transfection of Nrf2-siRNA or transient lentivirus infection containing Keap1 impedes neuronal differentiation [31].

NPCs derived from the adult mouse hippocampus provided further evidence of ORY’s effect in driving NPCs towards neuronal commitment, as demonstrated by increased MAP2-positive cells after ORY treatment, without affecting cell viability and apoptosis. This effect of ORY might again be attributed to the activation of Nrf2 pathways, since Murakami and colleagues [32] demonstrated that Nrf2 activity enhances both proliferation and differentiation of NPCs, and that the functional impairment of Nrf2 retards compensatory neurogenesis in a global brain ischemia model [32].

Interestingly, Yabe and colleagues [33] demonstrated that ferulic acid (FA), which is the major ORY derivative, increased the proliferation of rat telencephalon-derived NPCs without affecting the percentage of either neuron-specific class III β-tubulin-positive cells or GFAP-positive cells in the total cell population. They also discovered that oral ferulic acid boosted the number of new cells in the dentate gyrus of the hippocampus in corticosterone-treated mice, indicating its ability to enhance the proliferation of adult NSCs/NPCs in vivo [33]. Of note, the same authors reported that other natural antioxidant compounds, including baicalein, epigallocatechin gallate, and cinnamic acid, did not affect the proliferation of cultured NSCs/NPCs, suggesting that the promotion of NPC proliferation could be a selective feature of ferulic acid derivatives such as ORY.

In addition, evidence of the direct activation of the Nrf2–BDNF axis by ORY was supported by zebrafish experiments. In particular, our data demonstrated that ORY promotes neuronal differentiation and increases BDNF gene expression in Tg (-3.1*neurog1*:GFP) embryos. When Nrf2 genes were silenced with *nrf2a* and *nrf2b* morphants, ORY was unable to promote neurogenesis or transactivate BDNF.

Lastly, molecular docking experiments supported the strong binding affinity of the four ORY mixture compounds to the Kelch domain of human Keap1.

These findings suggest that once these compounds attach to the redox-sensitive domain of Keap1, their electrophilic nature, caused by the partial positive charge (δ+) of the olefin part, could lead to the oxidation of cysteine residues within the Kelch domain [34]. Interestingly, 24-methylenecycloartanyl ferulate, which is the most abundant in the ORY mixture, showed the most promising potential as an Nrf2 inducer. However, additional in vitro studies will be required to validate these findings.

## 5. Conclusions

In conclusion, our findings demonstrated that ORY can modulate neurite outgrowth and exert a neurotrophic effect, potentially involving the Nrf2 pathway. Additional studies will be needed to better clarify the connection between ORY, BDNF, and Nrf2–ARE signaling, and whether the observed effects are ascribable to ORY acting as a phytocomplex or to its individual components.

## Figures and Tables

**Figure 1 antioxidants-13-00969-f001:**
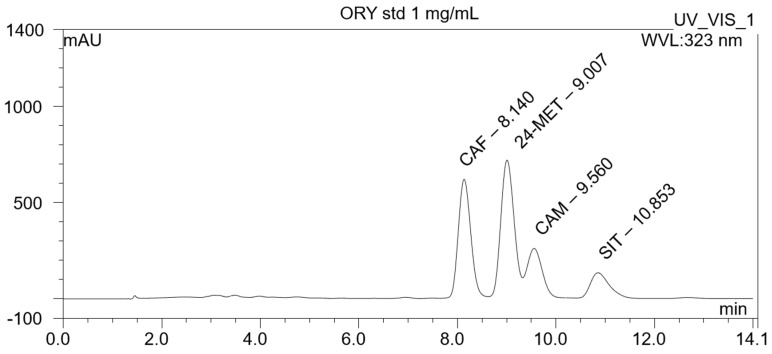
Chemical characterization of the ORY mixture. A list of the identified components is reported in the following: cycloartenyl ferulate (CAF, 37%), 24–methylenecycloartanyl ferulate (24–MET, 40%), campesteryl ferulate (CAM, 11%), and β–sitosteryl ferulates (SIT, 12%).

**Figure 2 antioxidants-13-00969-f002:**
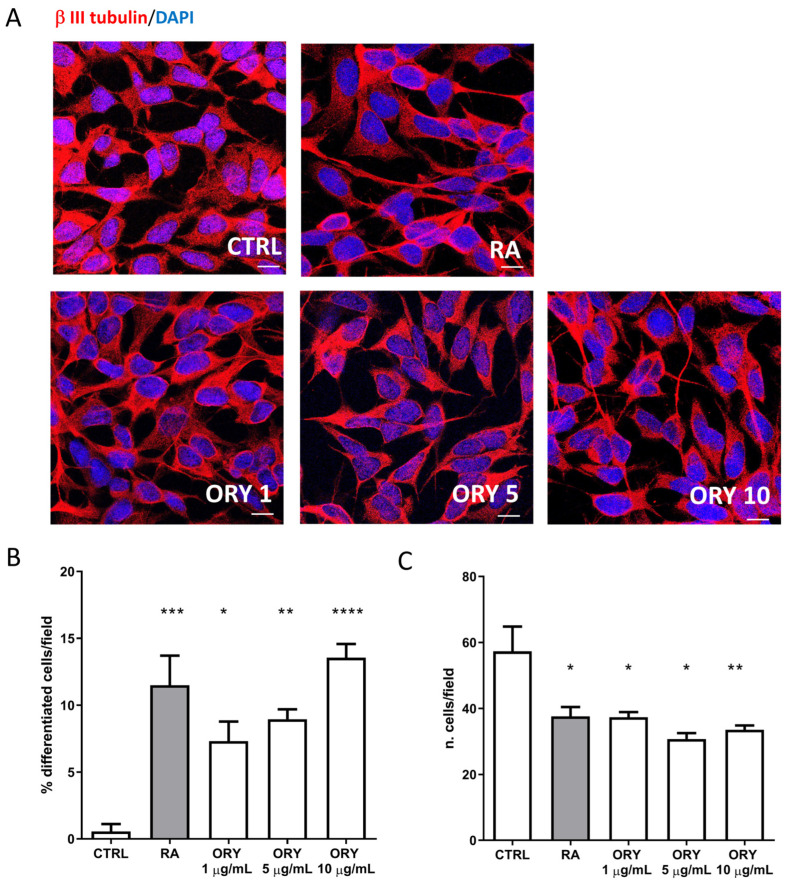
Gamma-oryzanol promotes neuronal differentiation in SH-SY5Y. SH-SY5Y is treated with a vehicle (CTRL), 10 µM retinoic acid (RA), or 1, 5, or 10 μg/mL gamma-oryzanol (ORY) for 5 days. (**A**) Representative confocal microscope images of SH-SY5Y labelled with β III-tubulin (red) and co-labelled with DAPI (blue). Magnification 63X, Scale bar = 10 µm. (**B**) The percentage of cells with neurites (**C**) and the number of cells per field were calculated in all the conditions respect to the control group. **** *p* < 0.0001; *** *p* < 0.001; ** *p* < 0.01; and * *p* < 0.05.

**Figure 3 antioxidants-13-00969-f003:**
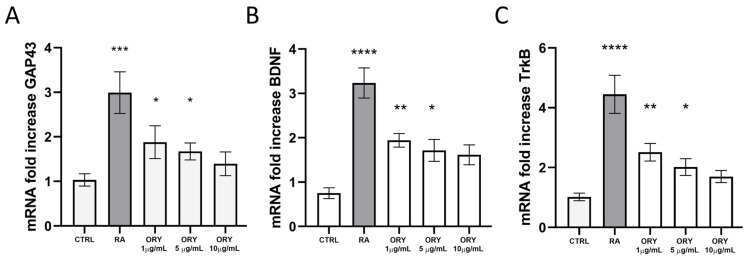
Gamma-oryzanol promotes neurotrophic effects in SH-SY5Y. Gene expression analysis of GAP43 (**A**), TrkB (**B**), and BDNF (**C**) normalized to the internal standard control gene (GAPDH). Data are represented as means ± S.E.M. of *n* = 3 experiments run in triplicate. One-way ANOVA tests with the Bonferroni post-test were used. **** *p* < 0.0001; *** *p* < 0.001; ** *p* < 0.01; and * *p* < 0.05.

**Figure 4 antioxidants-13-00969-f004:**
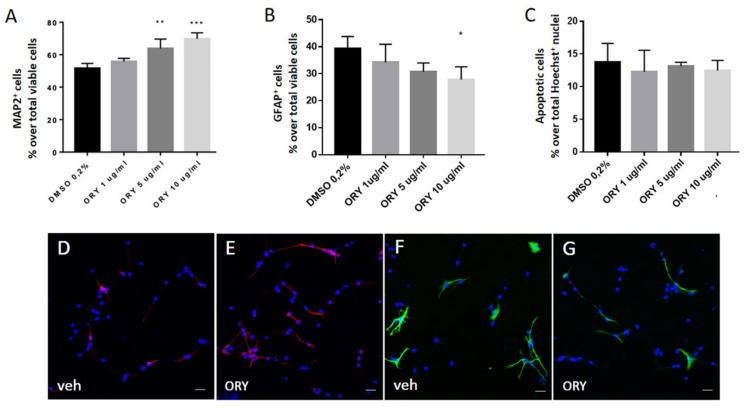
Gamma-oryzanol positively affects neuronal differentiation of ahNPCs. AhNPCs are seeded in differentiating conditions in the presence of a vehicle (DMSO 0.2%) or gamma-oryzanol (ORY) at 1, 5, or 10 μg/mL for 24 h. (**A**) Percentage of MAP2+ cells over total viable cells. (**B**) Percentage of GFAP+ cells over total viable cells. (**C**) Quantification of apoptotic cells calculated over the total number of Hoechst+ nuclei. (**D**–**G**) Representative confocal microscope images of MAP2+ (red, (**D**,**E**)) and GFAP+ (green, (**F**,**G**)) cells derived from ahNPCs differentiated in the presence of a vehicle (veh, (**D**,**F**)) or ORY (10 μg/mL, (**E**,**G**)). Nuclei are stained with Hoechst (blue). Magnification 63X, Scale bar = 20 µm. Data are represented as means ± SD of *n* = 3 experiments run in triplicate. Student’s *t*-test. *** *p* < 0.001; ** *p* < 0.01; * *p* < 0.05 vs. DMSO-treated cells.

**Figure 5 antioxidants-13-00969-f005:**
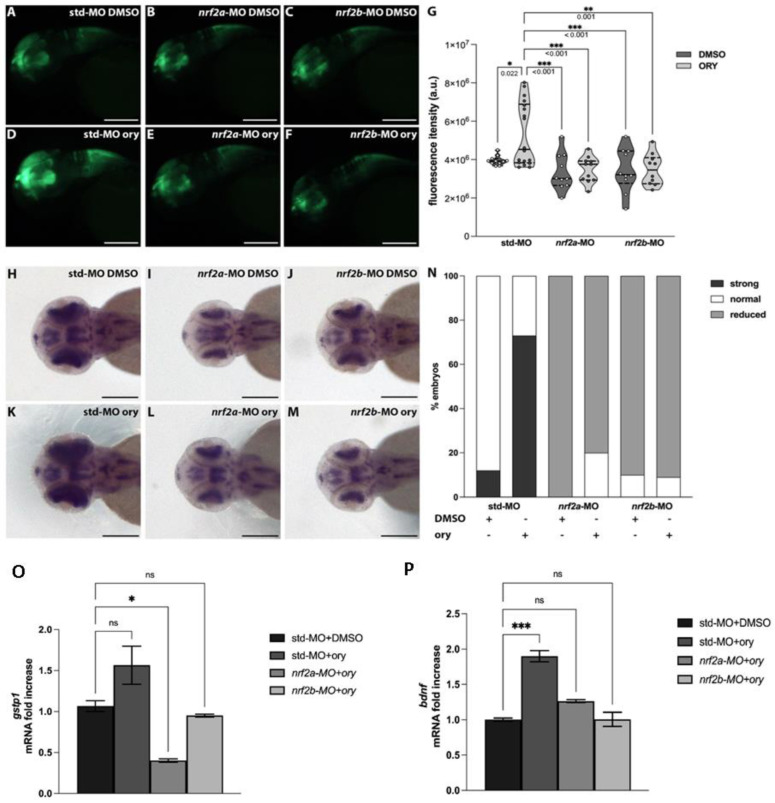
Gamma-oryzanol positively regulates nervous system development in zebrafish. (**A**–**F**) Representative images of the head region of 2 dpf Tg (-*3.1neurog1*:GFP) zebrafish embryos injected with std–MO or *nrf2*–MOs and then with DMSO or ORY: (**A**) std–MO + DMSO (*n* = 11); (**B**) *nrf2a*–MO + DMSO (*n* = 10); (**C**) *nrf2b*–MO + DMSO (*n* = 10); (**D**) std–MO + ORY (*n* = 18); (**E**) *nrf2a*–MO + ORY(*n* = 11); (**F**) *nrf2b*–MO + ORY (*n* = 10). (**G**) Quantification of the GFP fluorescence intensity of std–MO or *nrf2*–MOs embryos injected with DMSO or ORY. (**H**–**M**) *islet1* whole mount in situ hybridization of std–MO or *nrf2*–MOs embryos injected with DMSO or ORY: (**H**) std–MO + DMSO (*n* = 17); (**I**) *nrf2a*–MO + DMSO(*n* = 15); (**J**) *nrf2b*–MO + DMSO (*n* = 14); (**K**) std–MO + ORY (*n* = 15); (**L**) *nrf2a*–MO + ORY (*n* = 18); (**M**) *nrf2b*–MO + ORY (*n* = 11). (**N**) Subdivision of std–MO or *nrf2*–MOs embryos injected with DMSO or ORY according to the relative *islet1* expression. (**O**,**P**) RT-qPCR expression analyses of (**O**) *gstp1* and (**P**) *bdnf*. Scale bar indicates 100 μm. Data are presented as means ± standard error of the mean (S.E.M.). One–Way ANOVA with Tukey post–hoc. *** *p* < 0.001; ** *p* < 0.01; * *p* < 0.05; ns: not significant.

**Figure 6 antioxidants-13-00969-f006:**
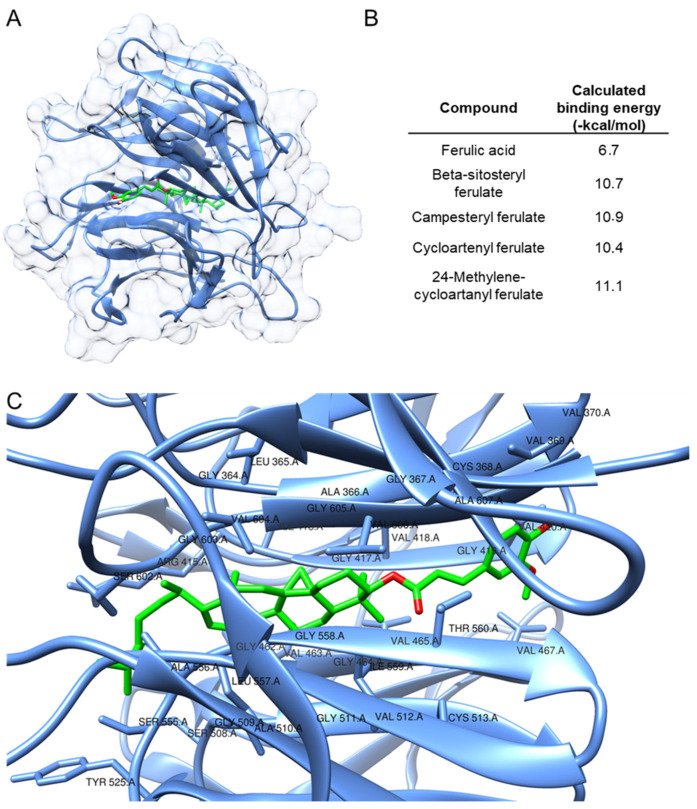
Interaction motif predicted by molecular docking for 24–methylenecycloartanyl ferulate (green), one of the components of ORY, with Keap1 (**A**): the compounds inserted below the hot spots of the Keap1 cavity (24–methylenecycloartanyl ferulate is depicted as a representative example). Calculated binding energy values retrieved for the studied compounds and ferulic acid were introduced as references (**B**). Detailed view of the binding site of Keap1: 24–methylenecycloartanyl ferulate is depicted in green (**C**). Interacting residues (<5 Å) have been labeled.

**Table 1 antioxidants-13-00969-t001:** List of primer sequences for real-time PCR. Gene accession number is included in square brackets.

Name	Primer Sequence
Growth-Associated Protein 43 (GAP-43) [NM_008083]	For: 5′-TTCTTGGTGTTGTTATGGCAA G-3′, Rev: 5′-GAGGAAAGTGGACTCCCACAG-3′
Brain-Derived Neurotrophic Factor (BDNF) [NM_001048141]	For: 5′-CATCCGAGGACAAGGTGGCTTG-3′, Rev: 5′-GCCGAACTTTCTGGTCCTCATC-3′
Tropomyosin receptor kinase B (TrkB) [NM_001025074.3]	For: 5′- GAGCATCATGTACAGGAAAT-3′, Rev: 5′- CTTGATGTTCTTCCTCATGT-3′,
Glyceraldehyde-3-Phosphate Dehydrogenase (GAPDH) [NM_008085]	For: 5′-GAGTCAACGGAT TTGGTCGT-3′, Rev: 5′-TTGATTTTGGAGGGATCTCG-3′.

## Data Availability

Data is contained within the article and Appendix A.

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
