# Peer review of "The Bioactive Gamma-Oryzanol from Oryza sativa L. Promotes Neuronal Differentiation in Different In Vitro and In Vivo Models"

_antioxidants, 2024, doi:10.3390/antiox13080969_

Round 1

Reviewer 1 Report

The authors demonstrated that gamma-oryzanol (ORY) from rice (Oryza sativa L.) consists predominantly of cycloartenyl, 24-methylenecycloartanyl, campesteryl, and sitosteryl ferulates, comprising about 80% of its content. Additionally, ORY exhibits antioxidant and anti-inflammatory properties. The authors employed both cell and zebrafish models to analyze the bioactive effects of ORY on neuronal differentiation. They discovered that ORY can promote neuronal differentiation in SH-SY5Y and mouse neural progenitor cells by modulating BDNF, TrkB, and GAP43 expression. Furthermore, using Nrf2 knockdown in zebrafish reduced ORY-induced Bdnf and gstp1 expression. However, several questions still need to be addressed.

Major concerns:

1.     Please refer to Figure 2A or B in the results section.

2.     Please display the differentiated cell phenotype, such as 𝛃III-tubulin length or intensity, to clarify the differences between differentiated cells and SH-SY5Y wild-type. Because Figures 2A and B are not clear enough to evaluate differentiated cells.

3.     Please use 24-methylenecycloartanyl ferulate or ferulic acid to investigate their effects on neuronal differentiation and gene expression (e.g., BDNF, TrkB, and GAP43).

The authors demonstrated that gamma-oryzanol (ORY) from rice (Oryza sativa L.) consists predominantly of cycloartenyl, 24-methylenecycloartanyl, campesteryl, and sitosteryl ferulates, comprising about 80% of its content. Additionally, ORY exhibits antioxidant and anti-inflammatory properties. The authors employed both cell and zebrafish models to analyze the bioactive effects of ORY on neuronal differentiation. They discovered that ORY can promote neuronal differentiation in SH-SY5Y and mouse neural progenitor cells by modulating BDNF, TrkB, and GAP43 expression. Furthermore, using Nrf2 knockdown in zebrafish reduced ORY-induced Bdnf and gstp1 expression. However, several questions still need to be addressed.

Major concerns:

1.     Please refer to Figure 2A or B in the results section.

2.     Please display the differentiated cell phenotype, such as 𝛃III-tubulin length or intensity, to clarify the differences between differentiated cells and SH-SY5Y wild-type. Because Figures 2A and B are not clear enough to evaluate differentiated cells.

3.     Please use 24-methylenecycloartanyl ferulate or ferulic acid to investigate their effects on neuronal differentiation and gene expression (e.g., BDNF, TrkB, and GAP43).

Reviewer 2 Report

The paper by dr. Abate and colleagues describes a pro-neuronal activity of the naturally occurring compound gamma-oryzanol, which would induce neuronal differentiation through Nrf2 activation and promotion of BDNF expression. Insights on possible interactions of ORY components with Keap1 are also provided. The work is well-organized and the results support the hypothesis.

In my opinion, a couple of points should be clarified. In addition, providing a few additional data (see below) may increase the interest toward this paper.

Lines 261 – 265: it should be noted that the highest ORY dose does not induce any significant increase in BDNF, TrkB, or GFAP43 expression (fig. 2D-F). In particular, while a dose-dependency effect of ORY could be observed favoring cell differentiation, it seems that an opposite dose dependency is present concerning BDNF, TrkB, and GFAP43 expression. This aspect should be discussed and possible explanations provided.

The experiments with zebrafish used intraventricular injections of 100 μg of ORY. How was this dose calculated? In the in vitro experiments, both with neuroblastoma cells and with hippocampal progenitors, the highest dose was 10 μg/ml. Therefore, the amount of ORY injected in the hindbrain ventricle of zebrafish seems to be much higher. The effect of ORY seems to be specific, since it disappears after blocking nrf2 expression, but some explanation about the choice of such a high dose (at least relative to that used in in vitro studies) should be provided. This would be necessary also considering that the highest dose of the in vitro experiments (10 μg/ml) was not effective in inducing BDNF expression, while this high dose used in zebrafish does stimulate BDNF expression.

The conclusion, supported by the results of this study and by other data in the literature, is that ORY (or one of its components) interacts with Keap1 and allows Nrf2 nuclear translocation and induction of gene expression resulting in BDNF increase and promotion of neuronal differentiation. I think the story would be more complete and the present work more “solid” if you could demonstrate Nrf2 nuclear translocation after treatment with ORY (a simple Nrf2 immunostaining would do it) in those neuroblastoma cells that are differentiated or in those ahNPCs that assume a neuronal phenotype. I know that you have demonstrated previously that ORY may induce Nrf2 nuclear translocation, but those data were in a different experimental model and not related to neuronal differentiation.

Minor:

Line 68: please define abbreviation NPCs

Line 69: please replace NCPs with NPCs

Lilne 236: there are no 3D structures depicted in supplementary figure 1.

Fig. 2A–C are not cited in the text. In fig. 2, retinoic acid is abbreviated as “AR”, while it should be “RA”, as indicated in the figure legend.

The names of the authors should be in the form of first name followed by surname (and not the opposite).

Round 2

Reviewer 1 Report

No further questions.

No further questions.

Reviewer 2 Report

I do not have any further comments. The revised version of the manuscript is fine with me.

I do not have any further comments. The revised version of the manuscript is fine with me.